# `zip2zip`: Inference-Time Adaptive Vocabularies for Language Models via Token Compression

**Saibo Geng** [* 1]   **Nathan Ranchin** [* 1]   **Yunzhen Yao** [1]   **Maxime Peyrard** [2]   **Chris Wendler** [1]   **Michael Gastpar** [1]   **Robert West** [1]

## Abstract

Tokenization efficiency plays a critical role in the performance and cost of large language models (LLMs), yet most models rely on static tokenizers optimized on general-purpose corpora. These tokenizers' fixed vocabularies often fail to adapt to domain- or language-specific inputs, leading to longer token sequences and higher computational costs. We introduce `zip2zip`, a framework that enables LLMs to dynamically adjust the token vocabulary at inference time, allowing for fewer generated tokens and thus faster inference. `zip2zip` consists of three key components: (1) a tokenizer based on Lempel-Ziv-Welch (LZW) compression that incrementally merges co-occurring tokens into reusable *hypertokens* on the fly; (2) an embedding layer that computes embeddings for newly formed hypertokens at runtime; and (3) a causal language modeling variant that trains the model to operate on hypertokenized, compressed sequences. We show that an existing LLM can be zip2zip-fied in 10 GPU-hours via parameter-efficient finetuning. The resulting zip2zip LLMs effectively learn to use hypertokens at inference time, reducing input and output sequence length by 20–60%, with significant improvements in inference latency. Code will be released at https://github.com/epfl-dlab/zip2zip.

## 1. Introduction

Large language models (LLMs) have shown impressive versatility across a broad spectrum of tasks and domains (Brown et al., 2020; Bubeck et al., 2023), including biomedical tests (Nori et al., 2023), mathematical reasoning (Frieder et al., 2023), programming (Jiang et al., 2024), and multiple human languages. A critical underlying component of this flexibility is the tokenizer, which defines the model's vocabulary and governs how raw text is converted into token sequence fed to the model. The efficiency of the tokenization scheme—i.e., how compactly a text is represented as tokens—has significant impact on model performance. In particular, a more *compact* tokenization yields three key benefits: (1) larger effective context window; (2) lower computational (and thus monetary) cost; and (3) shorter response times.

Despite its importance, the tokenizer used in most LLMs produces a fixed, static vocabulary using algorithms such as Byte Pair Encoding (Sennrich et al., 2016) over large-scale, general-purpose web corpora. While this globally optimized vocabulary performs reasonably well on average, it often fails to adapt to domain-specific or language-specific distributions (Ahia et al., 2023; Petrov et al., 2023), where the text distribution diverges significantly from the pretraining data. The resulting mismatch leads to longer token sequences, increasing both memory and compute demands, as well as the end user's cost by a factor of 2-3x when processing domain-specific text (Ahia et al., 2023). To mitigate this issue, prior work has explored expanding the token vocabulary during domain or language adaptation to improve tokenization efficiency (Wang et al., 2019; Zhao et al., 2024; Kim et al., 2024; Liu et al., 2023; 2024). While effective, this approach needs to be repeated for each target domain or language and requires maintaining separate tokenizers. Meanwhile, commercial LLM providers trend toward increasing the size of token vocabularies—growing from 32K to 128K (Grattafiori et al., 2024) and even up to 200K (Abdin et al., 2024) tokens—to improve overall tokenization efficiency. However, prior work (Dagan et al., 2024; Liang et al., 2023) shows that simply enlarging the vocabulary yields diminishing returns in domain adaptation, and vocabularies past a certain size can potentially degrade model performance (Liang et al., 2023).

These limitations point to a compelling need for an adaptive tokenization mechanism—one that can dynamically tailor the vocabulary to the input text at inference time, without retraining the model or maintaining separate tokenizers.

---

[*]Equal contribution  [1]EPFL  [2]Université Grenoble Alpes. Correspondence to: Saibo Geng <saibo.geng@epfl.ch>, Nathan Ranchin <nathan.ranchin@epfl.ch>, Robert West <robert.west@epfl.ch>.

*Non-archival presentation at ICML 2025 Tokenization Workshop (TokShop)*, Vancouver, Canada. 2025.

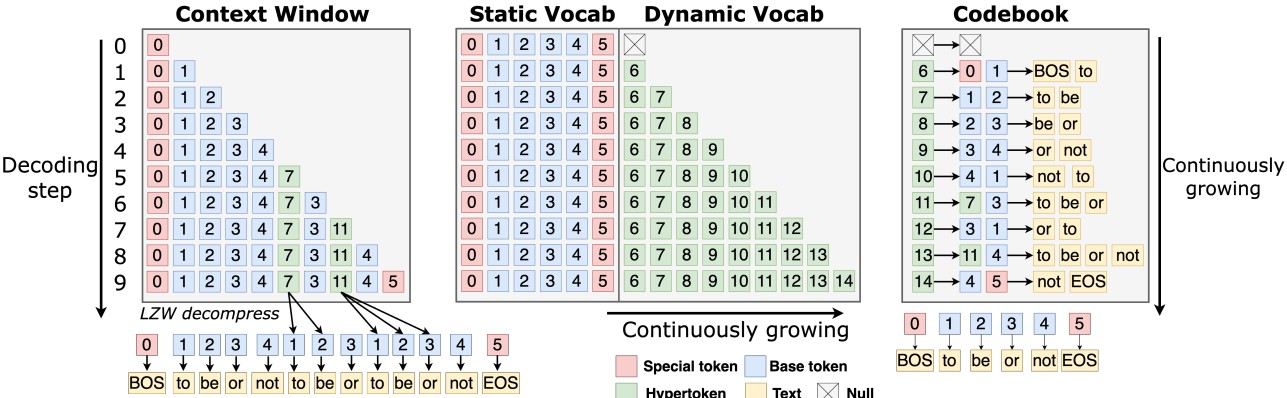

Figure 1: **Overview of the `zip2zip` inference pipeline.** At each decoding step, the model has a growing **context** composed of both **base tokens** (blue) and **hypertokens** (green). The **static vocabulary** of size 6 remains fixed, while the **dynamic vocabulary** is continuously expanded by merging co-occurring tokens using **LZW compression**. The **codebook** (right) maps hypertoken IDs to their corresponding base tokens. As decoding progresses, new hypertokens created at step $t$ (e.g., "to be", "or not") become **immediately** available for reuse at step $t+$. Additionally, output tokens, once generated, **instantly** become eligible for compression. Hypertokens are also eligible for merging, enabling the formation of **nested hypertokens**. The final output sequence (bottom) is reconstructed via LZW decompression.

Such a mechanism would allow the model to construct new domain-specific tokens on-the-fly, so to enhance tokenization efficiency. However, adaptive tokenization poses architectural challenges, as both the embedding layer and the language modeling head in transformer models (Vaswani et al., 2017) are static matrices tied to a fixed vocabulary size.

In this paper, we propose `zip2zip` (with a hat-tip to seq2seq (Sutskever et al., 2014)), a method that equips LLMs with a dynamic token vocabulary, enabling inference-time token adaptation. `zip2zip` achieves adaptive tokenization through *continuous vocabulary expansion* at runtime, allowing the model to represent a repeated or domain-specific pattern with a single long token rather than inefficient short tokens. This requires modest modifications to both the transformer architecture and the language modeling objective. `zip2zip` comprises three key components: (1) **Tokenizer:** an integration of Lempel-Ziv-Welch (LZW) compression[1] (Welch, 1984) into the tokenization process, which continuously merges frequently co-occurring token sequences into reusable longer tokens (hypertokens) at runtime; (2) **Architecture:** a lightweight encoder added to the transformer that computes embeddings for newly formed tokens on the fly; (3) **Training:** a compression-aware causal language modeling variant that trains the model directly on LZW-compressed sequences, aligning learning with the inference-time token distribution. The name `zip2zip` reflects its dual role in achieving compression of both the input tokens (the first *zip*) and output tokens (the second *zip*), thereby jointly improving the efficiency of input encoding and output decoding. We finetune `Phi-3-4B` and

Phi-3-14B to support `zip2zip` using as few as 100M tokens—requiring only 10 and 40 H100 GPU hours, respectively—for effective adaptation. The resulting models demonstrate strong inference-time compression capabilities and achieve 20–60% reductions in both input and output sequence lengths, translating to up to 60% improvements in end-to-end latency.

To make it easy to upgrade existing LLMs to `zip2zip`, we release an efficient, open-source implementation of the framework. It includes (1) a fast Rust-based LZW tokenizer, (2) a drop-in model architecture compatible with Hugging Face Transformers and vLLM, (3) a training pipeline for LZW-compression-based finetuning. Existing LLMs can be seamlessly extended with `zip2zip`, gaining adaptive tokenization capabilities through parameter-efficient finetuning—without any changes to the base model or tokenizer.

## 2. `zip2zip`

### 2.1. Dynamic Token Vocabulary

To enable dynamic tokenization at inference time, we associate each LLM with a *hyper-vocabulary* $\mathcal{V}_h$ that augments the model's static token vocabulary. Tokens from the original vocabulary $\mathcal{V}$ are referred to as *base tokens*. Each entry in the hyper-vocabulary is a *hypertoken*, representing a merged sequence of base tokens. The total vocabulary for a `zip2zip` model is the union $\mathcal{V} \cup \mathcal{V}_h$. At the beginning of each inference session, $\mathcal{V}_h$ is initialized as an empty set, and is incrementally populated during decoding by identifying and merging recurring token subsequences in the context window, as illustrated in Figure 1.

---

[1]LZW is the algorithm used in zip compression tool, which inspired the name `zip2zip`.

**Continuous Vocabulary Expansion.** As decoding proceeds, `zip2zip` continuously merge co-occurring tokens as new hypertokens to $\mathcal{V}_h$ and recurrently apply merging on newly generated tokens. This *continual expansion* allows the model to represent longer, recurring sequences of base tokens compactly. Hypertokens are treated as first-class tokens within the model, used interchangeably with base tokens throughout the decoding process. Importantly, this process occurs entirely during inference, without modifying the underlying tokenizer or requiring model retraining.

**LZW Algorithm.** We implement vocabulary expansion using the Lempel-Ziv-Welch (LZW) compression algorithm—a dictionary-based, lossless compression method that incrementally builds a codebook of variable-length sequences. In our setting, the codebook is initialized with the base token vocabulary $\mathcal{V}$ and expands by adding new hypertokens on the fly as recurring token patterns are encountered. To control the growth of the dynamically expanding vocabulary, we impose a maximum merge size $M$ that restricts how many base tokens a single hypertoken can represent. LZW is particularly well-suited for `zip2zip` due to the following properties:

(1) it is ***streaming***—hypertokens created at step $t$ can be immediately reusable at step $t + 1$; in contrast, methods like BPE require access to the full sequence and operate offline;

(2) it is ***self-contained***—input base tokens can be perfectly reconstructed from the compressed token sequence alone[2];

(3) it is ***unambiguous***—when both base tokens and hypertokens are available, which one to use is consistently determined by the LZW algorithm without ambiguity.

### 2.2. Hyper-Embedding and Hyper-Projection

Hypertokens do not have fixed embedding vectors in the original model's embedding layer (and projection layer), as they are not part of the original vocabulary. To compute the embedding of a hypertoken, we learn a mapping from the base token embeddings to the hypertoken embedding. We achieve this by introducing a *hyper-encoder*, which is a neural network that takes the embeddings of the constituent base tokens as input and outputs the corresponding hypertoken embedding. Specifically, for a sequence of $M$ base tokens $y_{1:M} := y_1 \ldots y_M$, the hyper-encoder $f_\phi : \mathcal{V}^M \to \mathbb{R}^d$ produces the hypertoken embedding $h = f_\phi(y_{1:M}) \in \mathbb{R}^d$, where $M$ is the maximum merge size and $d$ is the embedding dimension. For hypertokens composed of fewer than $M$ base tokens, we pad the input sequence to length $M$. Since the embedding map for base tokens remains unchanged, the hyper-encoder $f_\phi$ essentially maps the concatenated base

---

[2]There is no need to persist or transmit the codebook across inference calls, preserving compatibility with existing LLM libraries and interfaces.

token embeddings from a $(M \times d)$-dimensional space to a $d$-dimensional hypertoken embedding vector, performing nonlinear dimensionality reduction.

For the output projection layer, if the underlying transformer ties the embedding and the projection matrices, one can reuse the same hyper-encoder to compute the representation used for projection. Otherwise, a separate hyper-encoder is trained to produce the hypertoken projection vectors.

Let $\mathcal{V}$ be the original vocabulary of base tokens, with size $|\mathcal{V}|$, $\mathcal{V}_h$ be the set of hypertokens, with size $|\mathcal{V}_h|$, $E, P \in \mathbb{R}^{|\mathcal{V}| \times d}$ be the base token embedding/projection matrix. We define the augmented embedding matrix $\tilde{E} \in \mathbb{R}^{(|\mathcal{V}| + |\mathcal{V}_h|) \times d}$ and projection matrix $\tilde{P} \in \mathbb{R}^{(|\mathcal{V}| + |\mathcal{V}_h|) \times d}$ as:

$$\tilde{E} = \begin{bmatrix} E \\ H \end{bmatrix}, \quad \tilde{P} = \begin{bmatrix} P \\ P_h \end{bmatrix} \tag{1}$$

where:

- $H \in \mathbb{R}^{|\mathcal{V}_h| \times d}$ is the matrix of hypertoken embeddings, defined as

$$H = \begin{bmatrix} f_\phi(y_{1:M}^{(1)}) \\ \vdots \\ f_\phi(y_{1:M}^{(|\mathcal{V}_h|)}) \end{bmatrix} \in \mathbb{R}^{|\mathcal{V}_h| \times d} \tag{2}$$

- $P_h \in \mathbb{R}^{|\mathcal{V}_h| \times d}$ is the projection matrix of hypertokens, computed similarly with projection network $f_\psi$

### 2.3. Architecture

We illustrate the architecture of `zip2zip` in Figure 2. The input text is first tokenized into base tokens (**STEP 1**), which are then passed through an online LZW compressing module that compresses the token sequence into a stream of hypertokens (**STEP 2**). Since hypertokens are not part of the model's original embedding layer, their embedding vectors are computed on-the-fly using the *hyper-encoder* during inference (**STEP 3–4**). Once embedded, both base token embeddings and hypertokens embeddings are passed through the standard transformer layers of the base model, producing contextualized hidden states (**STEP 5–6**). This step is identical to vanilla transformer, with hypertokens and base tokens treated equally. At the output projection layer, hypertoken projection vectors (same as the hypertoken embedding vectors in the tied case, and computed by a separate hyper-encoder otherwise) are appended to the original projection matrix in the language modeling head (**STEP 7**). This allows the model to compute a joint logits over the union of the base vocabulary and the hyper vocabulary $\mathcal{V} \cup \mathcal{V}_h$ (**STEP 8**).

$$\text{logits}_t \in \mathbb{R}^{|\mathcal{V}| + |\mathcal{V}_h|} = h_t^\top \tilde{P}^\top = \begin{bmatrix} h_t^\top P^\top & h_t^\top P_h^\top \end{bmatrix} \tag{3}$$

where $h_t \in \mathbb{R}^d$ is the hidden state at timestep $t$.

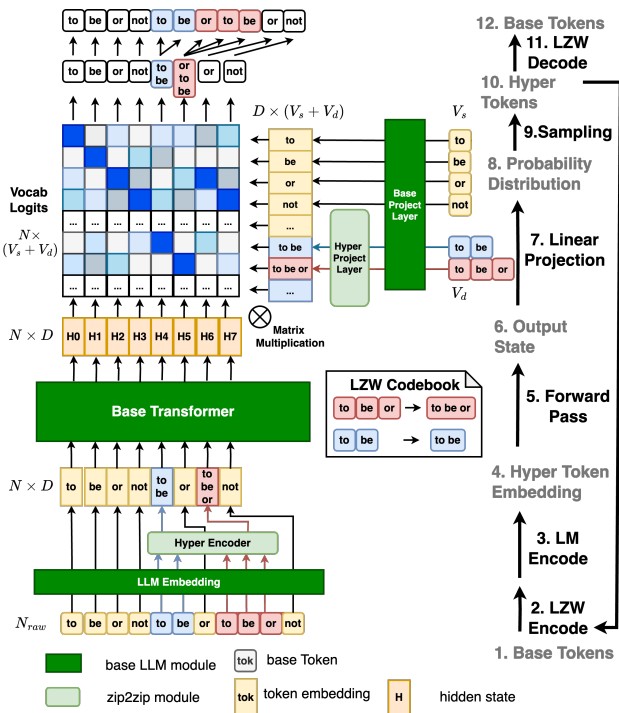

Figure 2: **zip2zip architecture.** At inference time, base tokens are compressed into hypertokens using LZW (**STEPS 1–2**). A hyper-encoder computes embeddings for hypertokens (**STEP 3–4**), which are processed by the base LLM (**STEPS 5–6**). Output representations are projected jointly on base and hyper-projection layers (**STEP 7**), producing joint logits and sampled tokens (**STEPS 8–10**), which can be decoded back to base tokens (**STEPS 11–12**).

The resulting probability distribution is over $\mathcal{V} \cup \mathcal{V}_h$, and the sampled token $\hat{z}_t$ may be either a base token or a hypertoken (**STEP 9**).

$$p_t \in \mathbb{R}_+^{|\mathcal{V}|+|\mathcal{V}_h|} = \text{softmax}(\text{logits}_t) \qquad (4)$$

$$\hat{z}_t \in \mathcal{V} \cup \mathcal{V}_h = \underset{i \in \{1,\dots,|\mathcal{V}\cup\mathcal{V}_h|\}}{\arg\max} \text{logits}_t[i] \qquad (5)$$

In the next cycle, the newly generated token $\hat{z}_t \in \mathcal{V} \cup \mathcal{V}_h$ (**STEP 10**) —whether base or hyper—is appended to the input sequence, and the process repeats (back to **STEP 1**).

At the end of generation, the hypertoken sequence is decompressed via the LZW decoding function into a sequence of base tokens (**STEP 11–12**). The whole process works in a fully *autoregressive* way, where newly generated tokens will also be merged into hypertokens for future steps. Furthermore, we highlight two points:

**Consistent Vocabulary Updates.** The expanding vocabulary—comprising newly created hypertokens—must be updated in a *consistent* manner across both the input embedding layer and the output projection layer, maintaining a consistent view of the hypertoken set. Failure to update both sides consistently can result in two types of errors:

(1) hypertokens that cannot be decoded, or (2) the model attempting to decode a non-existing hypertoken.

**Hyper-Embedding Cache.** Although hypertoken embeddings are computed on-the-fly, they are context-independent and can thus be cached across inference steps. Similar to the transformer's KV-cache, this enables *incremental* updates: only newly created hypertokens need to be processed at each step. Since the codebook grows linearly with the number of tokens in the context $n$, the total cache size grows also linearly in memory. Thus, the computational cost for hypertoken embeddings remains constant per step—i.e., one token embedding is computed per step.

### 2.4. Training `zip2zip` models

**Objective.** Let $\mathcal{D}$ denote the target text distribution. Given a language model $\pi_\theta$ parameterized by $\theta$, standard pretraining seeks to minimize the causal language modeling (CLM) objective, which corresponds to the expected negative log-probability of data sequences under the model:

$$\min_\theta \mathbb{E}_{y\sim\mathcal{D}} \left[ -\log \pi_\theta(y) \right], \qquad (6)$$

where $\pi_\theta(y)$ denotes the probability of the token sequence $y$ under the model $\pi_\theta$.

Let $\mathcal{C}$ be an *online* compression algorithm (e.g., LZW), and $\phi$ be the parameters of the hyper-encoder. Given a sequence $y \sim \mathcal{D}$, let $z = \mathcal{C}(y)$ be its compressed form. In `zip2zip`, we aim to optimize the same CLM loss, but over the compressed sequences $z$. The training objective becomes:

$$\min_{\theta,\phi} \mathbb{E}_{y\sim\mathcal{D}} \left[ -\log \pi_{\theta,\phi}(\mathcal{C}(y)) \right]$$
$$= \min_{\theta,\phi} \mathbb{E}_{z\sim\mathcal{C}(\mathcal{D})} \left[ -\log \pi_{\theta,\phi}(z) \right]. \qquad (7)$$

Here, we slightly abuse the notation to let $\pi_{\theta,\phi}(z)$ denote the probability assigned to the compressed sequence $z$, parameterized by the base model weights $\theta$ and the hyper-encoder parameters $\phi$.

To construct the compressed dataset $\mathcal{C}(\mathcal{D})$, we first tokenize the corpus using a standard tokenizer, and then apply the LZW compression algorithm. This preprocessing step is performed once prior to training and can be efficiently parallelized through batching.

**Parallelizable Training via Causal Masking.** Although hypertokens introduce additional vocabulary dynamics, training remains fully parallelizable. We leverage the standard causal masking mechanism used in language models, allowing the model to predict the next token—whether a base token or a hypertoken—at each position in parallel. To eliminate the need for sequential codebook updates during inference, we precompute a fixed codebook by applying

LZW compression to the entire input sequence. This pre-computed codebook is then used consistently throughout training to condition token predictions, ensuring efficiency and compatibility with standard training pipelines.

**Auxiliary Reconstruction Loss.** We introduce an auxiliary reconstruction objective that encourages a hypertoken embedding to retain sufficient information about its underlying base token sequence. Specifically, the model is trained to reconstruct the original base token embeddings from the hypertoken embedding. We jointly optimize the language model and the hyper-encoder using a combined loss that includes both the standard next-token prediction loss and the auxiliary reconstruction loss. Formally, we optimize:

$$\min_{\theta,\phi,\psi} \mathbb{E}_{y \sim \mathcal{D}} \left[ - \log \pi_{\theta,\phi}(\mathcal{C}(y)) \right]$$
$$+ \lambda \, \mathbb{E}_{y_{1:M}} \left[ \Delta \left( y_{1:M}, f_\psi \left( f_\phi(y_{1:M}) \right) \right) \right], \tag{8}$$

where $f_\phi : \mathcal{V}^M \to \mathbb{R}^d$ is the hyper-encoder, $f_\psi : \mathbb{R}^d \to \mathcal{V}^M$ is the decoder aiming to reconstruct the corresponding base tokens from their hyper-embedding, and $\Delta : \mathcal{V}^M \times \mathcal{V}^M \to \mathbb{R}$ is the reconstruction loss function, such as the cross-entropy loss, between the base tokens $y_{1:M}$ and the reconstructed base tokens $f_\psi \left( f_\phi(y_{1:M}) \right)$. The hyperparameter $\lambda$ controls the trade-off between the prediction error of the language model and the reconstruction error of the autoencoder. This joint optimization objective encourages the hyper-encoder to learn a compact $d$-dimensional manifold embedded in the higher-dimensional $(M \times d)$ space of base token embeddings, while the language model $\pi_{\theta,\phi}$ learns to predict the next (hyper)token given the preceding context. The reconstruction loss can be viewed as a form of auto-encoding, where the hypertoken acts as a compressed latent representation and reconstruction encourages the preservation of semantic content and the compression to be lossless.

**Adapting Pretrained Language Models.** The proposed objectives (Equation 7, 8) integrate naturally with pretrained language models. In this setting, the base model can be frozen while training only the hyper-encoder to adapt to compressed token sequences. Parameter-efficient methods such as LoRA (Hu et al., 2022) may also be used to adapt select components of the base model, enabling effective adaption with minimal computes.

## 2.5. Efficiency Advantage

zip2zip improves efficiency by increasing the average token length, thereby reducing the number of tokens required to represent the same text. This compression applies to both inputs (e.g., prompts) and outputs (e.g., completions), leading to shorter effective context lengths. As a result, the model performs fewer computations—both in the attention mechanism and the feedforward layers—and, more importantly, requires fewer autoregressive decoding steps during

inference. Since the latency of large language models is primarily driven by the cost of sequential decoding, reducing the number of output tokens by $n\%$ leads to an approximate $n\%$ speedup in decoding latency, which we will demonstrate empirically in Section 3.6. A more detailed discussion of FLOPs is provided in Appendix B for completeness.

## 3. Experiments

To evaluate the effectiveness of zip2zip, we adapt the Phi-3 models (3B and 14B) within the zip2zip framework. We evaluate our adapted models across four dimensions: (1) token efficiency, (2) language modeling perplexity, (3) downstream task performance, (4) inference efficiency. For perplexity and downstream benchmarks, we use the widely adopted lm-evaluation-harness framework (Gao et al., 2024).

### 3.1. Training Setup

Rather than updating the full model weights, we adopt parameter-efficient finetuning using LoRA (Hu et al., 2022). In addition, we train the *hyper-embedding* and *hyper-projection* modules. We set the maximum merge size to $M = 3$ and use a two-layer transformer encoder as the hyper-encoder. Ablation studies on $M$ and hyper-encoder architecture can be found in Appendix A. For comparison, we also perform continual finetuning of the base model using LoRA under identical training conditions, serving as a baseline (denoted as Cont. Finetune in the Tables) The finetuning process is highly efficient, requiring approximately 10 H100-GPU hours for a 4B-parameter model and up to 40 H100-GPU hours for a 14B-parameter model, using only 0.1 billion training tokens. Interestingly, the *reconstruction loss* converges to **near zero** during training, indicating that the model can almost perfectly recover the original base token sequences from the hypertoken representations. This highlights the learned compression is highly information-preserving. Details of the training setup, compute infrastructure, and dataset curation are provided in Appendices D and E.

### 3.2. Sample Outputs and Hypertoken Patterns

We present several examples to provide intuition into how the zip2zip model generates text.

We see that the model successfully generates a mixture of hypertokens and base tokens in the output (see Figure 3). The hypertoken ratio is as high as 40% in the Python code generation example, and 20% in the biomedical text generation example. Many of the hypertokens correspond to semantically meaningful units or domain-specific terms as shown in Table 1. For a more fine-grained visualization

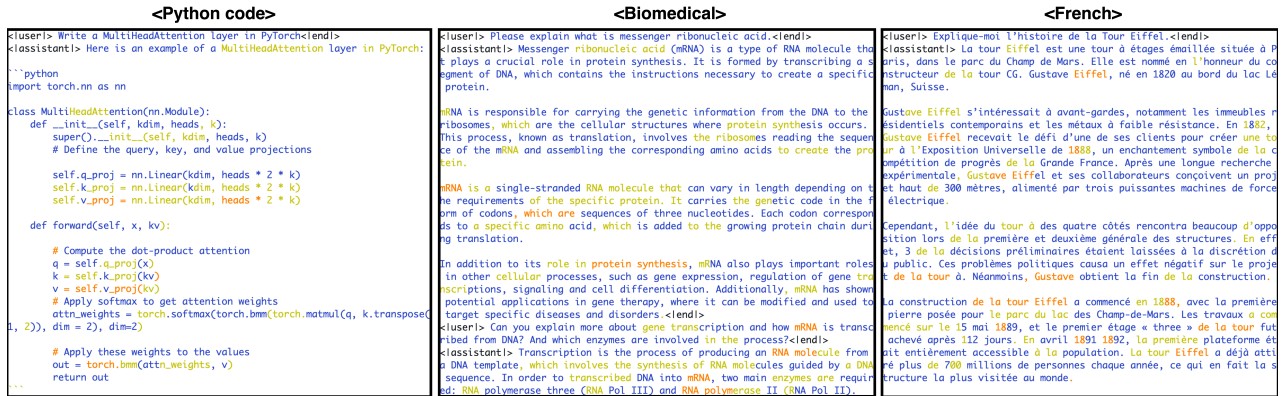

Figure 3: **Zip2Zip output examples.** Blue: base tokens; Yellow: hypertokens (composed of 2 base tokens); Orange: hypertokens (composed of 3+ base tokens).

Table 1: Examples of hypertokens formed by zip2zip across three domains

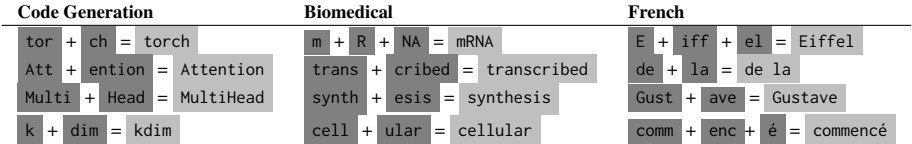

of hypertoken with `zip2zip`, we provide visualizations of token streams in Appendix 8.

### 3.3. Token Efficiency

Given an input text $x$ and a tokenizer, we define *token efficiency* $\eta := \frac{\text{Bytes}(x)}{\text{Tokens}(x)}$ as the average number of bytes represented by each token, where $\text{Bytes}(x)$ refers to the number of bytes in the UTF-8 encoding of $x$. This measures how compactly a tokenizer encodes input text—higher values of $\eta$ indicate more efficient tokenization.

We evaluate token efficiency using the tokenizers of four LLMs—Llama-3 (Grattafiori et al., 2024), Qwen-2 (Yang et al., 2024), Phi-4 (Abdin et al., 2024), and Gemma-3 (Team et al., 2025)—each associated with a different base vocabulary size ranging from 128K to 256K. Token efficiency is measured across five representative domains, sampled from publicly available datasets: code (Lozhkov et al., 2024b), math (LI et al., 2024), chat (Ding et al., 2023), multilingual (Penedo et al., 2024), and web (Lozhkov et al., 2024a). Table 2 shows that applying LZW `zip2zip` consistently improves token efficiency across all tokenizer and domains. Gains are particularly strong in structured domains like code and math—50% higher than the base tokenizer. Interestingly, models with larger vocabulary sizes do not always achieve better token efficiency, suggesting that simply enlarging the vocabulary size is not sufficient to improve it.

### 3.4. Perplexity

We evaluate the perplexity of `zip2zip` models on four corpora: Wikitext (Merity et al., 2016), the Pile (Gao et al., 2020), and two subsets of Paloma (Magnusson et al., 2023): mC4, a multilingual subset of C4, and dC4 (aka C4-100D), a subset of C4 spanning 100 domains. Given a token sequence $x = x_1, \ldots, x_N$, and a model $q$, perplexity and byte-level perplexity (Radford et al., 2019; Magnusson et al., 2023) are defined as: $\text{PPL} := \left( \prod_{i=1}^{N} q(x_i) \right)^{-1/N}$, $\text{Byte-PPL} := \left( \prod_{i=1}^{N} q(x_i) \right)^{-1/B} = \text{PPL}^{1/\eta}$, where $B$ is the number of UTF-8 bytes of the text, and $\eta$ denotes the token efficiency (i.e., bytes per token). Token-level perplexity depends on the tokenization scheme and is unsuitable for cross-tokenizer comparison. We instead report byte-level perplexity, a vocabulary-agnostic metric that normalizes for tokenization differences. Table 3 shows that `zip2zip` model has a modest increase in Byte-perplexity, indicating a drop in language modeling performance.

### 3.5. Evaluation on NLP Benchmarks

We next evaluate `zip2zip`'s performance on real-world tasks. We evaluate on seven widely used NLP benchmarks, including ARC-[Challenge, Easy] (Clark et al., 2018), HellaSwag (Zellers et al., 2019), LAMBADA (Paperno et al., 2016), OpenbookQA (Mihaylov et al., 2018), PIQA (Bisk et al., 2019), Winogrande (Sakaguchi et al., 2019) and GSM8K (Cobbe et al., 2021). As shown in Table 4, the model finetuned with `zip2zip` performs similarly to the

Table 2: **Token efficiency** (bytes/token) across domains for different tokenizers w/wo `zip2zip`.

| Tokenizer | Code | Math | Chat | Multilingual | Web |
|---|---|---|---|---|---|
| Llama-3-128K (Grattafiori et al., 2024) | 4.1 | 2.7 | 5.1 | 3.8 | 4.6 |
| +zip2zip | 6.3 (+54%) | 4.0 (+48%) | 6.4 (+25%) | 4.7 (+24%) | 5.4 (+17%) |
| Qwen-2-150K (Yang et al., 2024) | 4.0 | 2.3 | 5.1 | 3.7 | 4.4 |
| +zip2zip | 6.2 (+55%) | 3.7 (+61%) | 6.4 (+25%) | 4.6 (+24%) | 5.2 (+18%) |
| Phi-4-200K (Abdin et al., 2024) | 4.1 | 2.7 | 5.4 | 4.6 | 4.7 |
| +zip2zip | 6.3 (+54%) | 4.1 (+52%) | 6.7 (+24%) | 5.5 (+20%) | 5.4 (+15%) |
| Gemma-3-256K (Team et al., 2025) | 3.3 | 2.3 | 5.0 | 4.4 | 4.5 |
| +zip2zip | 5.6 (+70%) | 3.7 (+61%) | 6.4 (+28%) | 5.4 (+23%) | 5.4 (+20%) |

Table 3: Byte-perplexity ($\downarrow$) on four corpora using a 1024-token context window.

| Model | Method | Wiki | Pile | mC4 | dC4 |
|---|---|---|---|---|---|
| Phi-3.5-4B | Base | 1.62 | 1.88 | 1.94 | 1.77 |
| | Cont. finetune | 1.63 | 1.89 | 1.94 | 1.77 |
| | zip2zip | 1.71 | 2.02 | 2.04 | 1.84 |
| Phi-3-14B | Base | 1.43 | 1.72 | 1.82 | 1.67 |
| | Cont. finetune | 1.47 | 1.79 | 1.86 | 1.68 |
| | zip2zip | 1.56 | 1.90 | 1.96 | 1.75 |

Table 4: Two-shot accuracy (in %) across 7 NLP benchmarks. Higher is better. Standard deviations (bootstrapped) $\approx 0.02$ across all tasks. C.F.=Continuous finetune, Z2Z=zip2zip.

| Benchmark | Phi-3.5-4B | | | Phi-3-14B | | |
|---|---|---|---|---|---|---|
| | Base | C.F. | Z2Z | Base | C.F. | Z2Z |
| ARC-c | 0.60 | 0.60 | 0.57 | 0.62 | 0.62 | 0.62 |
| ARC-e | 0.83 | 0.82 | 0.83 | 0.80 | 0.88 | 0.86 |
| HS | 0.66 | 0.63 | 0.61 | 0.70 | 0.66 | 0.68 |
| OBQA | 0.46 | 0.47 | 0.46 | 0.51 | 0.52 | 0.51 |
| PIQA | 0.79 | 0.82 | 0.82 | 0.83 | 0.87 | 0.85 |
| WG | 0.75 | 0.75 | 0.75 | 0.76 | 0.80 | 0.79 |
| GSM8K | 0.82 | 0.40 | 0.15 | 0.84 | 0.52 | 0.25 |

baseline on most tasks. However, on GSM8K, where the primary task involves numerical computation, the model exhibits significant degradation. Due to the sensitivity of such tasks to tokenization, it occasionally generates malformed or repeated numbers. While token-level operations are already known to be challenging for LLMs (Singh & Strouse, 2024), adaptive tokenization appears to exacerbate this issue.

To validate the effectiveness of `zip2zip` on non-English languages, we evaluate the model on machine translation tasks, including WMT14 (Macháček & Bojar, 2014), WMT16 (Bojar et al., 2016). The results, shown in Table 5, indicate a small performance degradation across BLEU, CHRF, and TER metrics when using `zip2zip`. However, the drop is relatively minor, suggesting that the model re-

tains strong multilingual capabilities even in the compressed representation.

Table 5: Machine translation performance on WMT benchmarks. Scores are averaged across both translation directions. Standard deviations (approximately $1.0 \sim 2.0$) are reported in Table 10 in Appendix C.

| Model | Method | Metric | WMT14 En-Fr | WMT16 En-De | WMT16 En-Ro |
|---|---|---|---|---|---|
| Phi-3.5-4B | Base | BLEU↑ | 33.6 | 39.2 | 17.7 |
| | | CHRF↑ | 58.3 | 63.2 | 45.5 |
| | | TER↓ | 53.0 | 47.9 | 73.4 |
| | Cont. finetune | BLEU↑ | 36.5 | 42.3 | 16.7 |
| | | CHRF↑ | 61.0 | 65.4 | 45.8 |
| | | TER↓ | 51.5 | 44.9 | 79.7 |
| | zip2zip | BLEU↑ | 34.1 | 39.7 | 14.3 |
| | | CHRF↑ | 59.4 | 64.5 | 44.2 |
| | | TER↓ | 54.5 | 48.0 | 93.5 |
| Phi-3-14B | Base | BLEU↑ | 39.1 | 43.1 | 21.3 |
| | | CHRF↑ | 62.6 | 65.6 | 51.0 |
| | | TER↓ | 49.3 | 44.1 | 70.5 |
| | Cont. finetune | BLEU↑ | 38.9 | 48.4 | 21.8 |
| | | CHRF↑ | 63.2 | 70.1 | 52.0 |
| | | TER↓ | 48.8 | 39.8 | 68.3 |
| | zip2zip | BLEU↑ | 36.4 | 44.8 | 19.5 |
| | | CHRF↑ | 62.8 | 68.1 | 50.1 |
| | | TER↓ | 51.2 | 42.9 | 72.9 |

### 3.6. Inference Efficiency

`zip2zip` reduces decoding time by lowering the number of tokens that need to be generated. However, it introduces additional FLOPs due to the on-the-fly computation of hyper-embeddings by the hyper-encoder. To address this overhead, we implement hyper-embedding caching and optimize the computation using a custom Triton kernel. We report separate timings for *prefill* and *decoding* across multiple models, with and without `zip2zip`, in Table 6.

As we show in Table 6, `zip2zip` achieves a significant speedup in all four settings. Both prefill and decoding times are significantly reduced, with the most substantial gains observed in the 512+256 setting with the Phi-3.5-4B model.

Table 6: **Throughput (tokens/sec)** comparison of the `zip2zip` framework against the baseline Hugging Face Transformers `generate` and MLX `generate` implementation. Performance is detailed for prefill and decode phases across various context lengths (first value in column headers) combined with a 256-token generation length. `zip2zip` demonstrates notable throughput improvements, in both prefill and decoding phase.

| Setting | Method | 256+256 | | 512+256 | | 1024+256 | | 2048+256 | |
|---|---|---|---|---|---|---|---|---|---|
| | | **Prefill** | **Decode** | **Prefill** | **Decode** | **Prefill** | **Decode** | **Prefill** | **Decode** |
| *Hardware: Apple M1 (16GB RAM)* | | | | | | | | | |
| **Phi-3-4B** | Base model | 165.0 | 7.3 | 211.3 | 7.5 | 200.9 | 7.1 | 196.6 | 6.8 |
| | zip2zip | 145.5 | 7.9 | 231.4 | 10.1 | 189.6 | 7.4 | 233.8 | 7.3 |
| | **Relative %** | -11.8% | +7.5% | +9.5% | +34.8% | -6.6% | +3.9% | +18.9% | +7.5% |
| *Hardware: NVIDIA H100 80GB GPU* | | | | | | | | | |
| **Phi-3.5-4B** | Base model | 700.9 | 56.2 | 1347.2 | 54.4 | 2689.4 | 52.8 | 4993.2 | 53.1 |
| | zip2zip | 936.6 | 61.4 | 2722.1 | 79.8 | 4326.7 | 61.5 | 9258.1 | 61.9 |
| | **Relative %** | +33.6% | +9.3% | +102.6% | +46.6% | +60.9% | +16.6% | +85.4% | +16.5% |
| **Phi-3-14B** | Base model | 724.4 | 44.6 | 1356.3 | 43.8 | 2328.6 | 45.1 | 3849.5 | 42.2 |
| | zip2zip | 1024.6 | 54.9 | 1973.0 | 61.1 | 3657.0 | 66.8 | 7239.1 | 46.3 |
| | **Relative %** | +41.5% | +23.0% | +45.5% | +39.5% | +57.0% | +48.1% | +88.1% | +9.6% |

Improvements are significantly stronger on datacenter-grade GPUs like the NVIDIA H100 and more modest on consumer hardware (e.g., Apple M1).

**Efficient LZW Tokenization.** `zip2zip` introduces an additional LZW compression step during inference and a de-compression step at the end of generation. As a result, the efficiency of LZW-integrated tokenization is important to overall performance. To minimize overhead, we implemented a Rust-based `zip2zip` tokenizer that outperforms the Python version (see Figure 4) and matches the latency of HuggingFace's fast BPE tokenizer.

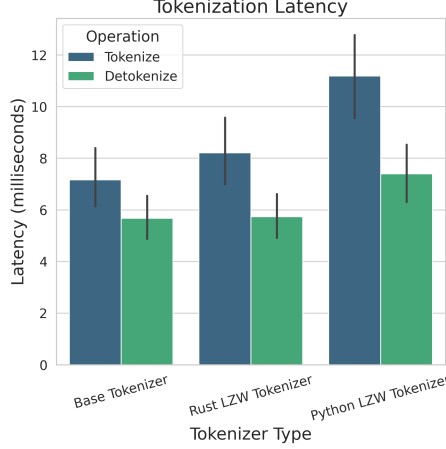

Figure 4: `zip2zip` tokenizer latency (ms) vs. HF tokenizer.

## 4. Related Work

**Vocabulary Expansion.** Several works have explored expanding the tokenizer vocabulary to better support specific domains or languages. Zhao et al. (2024); Kim et al. (2024); Liu et al. (2023; 2024) adapt LLaMA to Chinese, Korean, and specialized domains such as mental health and law by appending new tokens. Wang et al. (2025); Liu et al. (2024) conducted studies on how to effectively expand the vocabulary by better selecting the subset of tokens to add. In contrast, `zip2zip` is the first to enable *dynamic vocabulary expansion at inference time*, constructing new tokens based on the input context without requiring retraining or modifying the tokenizer ahead of time.

**Prompt Compression.** Prompt compression methods include GistTokens (Mu et al., 2023), Selective Context (Li et al., 2023), LLMLingua (Jiang et al., 2023), Summary Vectors (Chevalier et al., 2023), In-context Autoencoder (Ge et al., 2024), and others (Wingate et al., 2022) reduce the input token length and but do not impact the number of output tokens, which often dominates overall generation time. In contrast, `zip2zip` compresses *both* the input and output token sequences.

**Latent Tokens Representation.** The concept of latent token representations, or *patches*, has been mostly explored in computer vision, with methods like Token Merging (Bolya et al., 2023) and Token Pooling (Marin et al., 2023) aiming to reduce sequence length while preserving semantic content. Recently, Byte Latent Transformer (BLT) (Pagnoni et al., 2024) extended this concept to language modeling by discarding tokens entirely and operating directly at the byte level. Both BLT and `zip2zip` adopt a hierarchical modeling of input for LLMs, but they differ in three key

ways: (1) **Goal**: BLT aims to replace the tokenizer, whereas `zip2zip` seeks to expand and improve it; (2) **Algorithm**: BLT uses entropy-based segmentation, while `zip2zip` applies LZW-based token compression; (3) **Training**: BLT requires training from scratch, whereas `zip2zip` enables continued adaptation of pretrained models. Lester et al. (2024) propose improving language model efficiency by training LLMs directly on text compressed with arithmetic coding. While both approaches leverage compression to enhance efficiency, `zip2zip` emphasizes dynamic vocabulary expansion to enable uptraining of existing models. In contrast, Lester et al. (2024) requires training from scratch.

## 5. Discussion and Limitations

**Beyond LZW.** While we adopt LZW for dynamic construction of hypertokens, `zip2zip` is broadly compatible with any online compression algorithm. Future work may explore alternative schemes that provide different trade-offs between compression efficiency and model performance.

**Codebook Management Strategy.** The LZW algorithm grows the codebook linearly with the number of tokens in the context window. Empirical results show that only about 25% of hypertokens are reused during generation, leaving substantial room for optimization. Two potential improvements are: (1) **pruning** or **selective retention** strategies to reduce unused entries, and (2) **codebook prefilling**, which could be beneficial if likely tokens can be anticipated before input processing.

**Compression–Quality Trade-off.** There is an inherent trade-off between compression and modeling: as the token space is compressed more aggressively, redundancy is reduced—but so is predictability—making it harder for the model to forecast the next (hyper)token. In the extreme, optimal compression schemes such as arithmetic coding produce sequences that are statistically indistinguishable from random noise, rendering them unlearnable by language models (Lester et al., 2024). Empirically, we observe this effect as increased perplexity under higher compression levels (Table 3), which can undermine the benefits of compression by degrading generation quality (though minor in the tasks in Table 4 and Table 5). Striking the right balance between compression and model performance remains an important direction for future research.

## 6. Conclusion

We introduced `zip2zip`, a framework for inference-time vocabulary adaptation with LLMs. By integrating LZW-based token compression with a dynamic hypertoken embedding mechanism, `zip2zip` enables substantial reductions in sequence length and decoding steps, leading to improved inference efficiency with minimal architectural modifications.

Our experiments demonstrate that `zip2zip` maintains strong performance across a range of tasks while achieving significant gains in inference efficiency. These findings highlight the promise of integrating dynamic tokenization into LLMs, opening up new directions for research in LLM efficiency.

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

# A. Ablation Studies

**Definition A.1** (Compression Rate). *We define the* compression rate *as the ratio between the number of tokens after compression ($N_{comp}$) and the number of tokens in the original uncompressed text ($N_{orig}$), expressed as a percentage:*

$$Compression\ Rate = \frac{N_{comp}}{N_{orig}} \times 100\%.$$

*A lower compression rate indicates greater reduction in token count, and thus more effective compression.*

## A.1. LZW Maximum Merge Size

The last column of Table 7 shows how the maximum merge size $M$ affects compression rate when the context window length is 2048. As $M$ increases, compression rate improves significantly, especially from $M = 1$ to $M = 3$. Beyond that, gains diminish, suggesting $M = 3$ strikes a good balance between efficiency and compression rate.

Interestingly, the relationship between maximum merge size and training loss in Figure 5 as well as perplexity in Table 7 is non-monotonic. The baseline case with $M = 1$ (i.e., no zip2zip compression) yields the lowest perplexity overall, which is expected and consistent with prior findings that compression typically incurs a trade-off in model performance. Among the compressed settings, the case $M = 2$ performs the worst, with noticeably slower convergence and higher final loss. In contrast, the case $M = 3$ achieves the best performance within the compressed configurations, striking a favorable balance between compression and prediction performance. While $M = 4$ and $M = 5$ also perform reasonably well, they exhibit slightly higher loss than $M = 3$, suggesting diminishing returns or possible over-compression at larger maximum merge sizes (see Figure 5).

Table 7: **Effect of maximum merge size** ($M$) **on byte-level perplexity and compression rate.** Perplexity is measured for Phi-3.5-4B across four corpora with a 1024-token context window. Compression rate is evaluated over the training corpus with a 2048-token context. $M = 1$ corresponds to no compression.

| $M$ | Wiki | Pile | mC4 | dC4 | Compression Rate(%) |
|---|---|---|---|---|---|
| 1 | 1.62 | 1.70 | 2.00 | 1.91 | 100.00 |
| 2 | 1.96 | 2.21 | 2.55 | 2.22 | 75.30 |
| 3 | 1.72 | 1.84 | 2.15 | 2.00 | 71.21 |
| 4 | 1.71 | 1.84 | 2.14 | 1.99 | 68.93 |
| 5 | 1.72 | 1.84 | 2.14 | 1.99 | 68.41 |

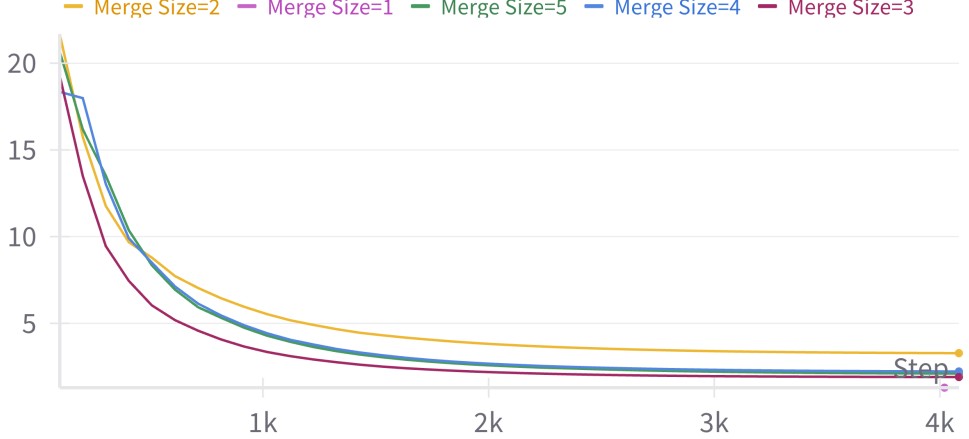

Figure 5: **Effect of maximum merge size $M$ on zip2zip training loss**: $M = 1$ (no compression) achieves the lowest loss overall. Among compressed settings, $M = 3$ performs best, while $M = 2$ shows the worst convergence. Larger $M$ (4 and 5) yield slightly worse results than $M = 3$.

Table 7 reports the byte-level perplexity across four corpora using a 1024-token context window. The results align closely with the training loss trends observed earlier. Setting $M = 1$ (i.e., no compression) consistently achieves the lowest perplexity across all datasets, reaffirming that compression introduces a performance trade-off. Notably, $M = 2$ performs the worst across all corpora, exhibiting the highest perplexity values. For merge sizes $M = 3$, $M = 4$, and $M = 5$, perplexity scores are nearly identical, suggesting that moderate compression can be achieved without significantly sacrificing

language modeling quality—provided $M = 2$ is avoided. This consistency across loss and perplexity metrics further supports the robustness of maximum merge size $M = 3$ as the most effective trade-off point.

### A.2. Hyper-encoder architecture

We ablate the architecture of the hyper-encoder to evaluate its effect on language modeling performance, as shown in Table 8. We compare increasingly expressive architectures, starting with a simple averaging method that introduces no additional parameters. This baseline yields the highest perplexity, highlighting its limited capacity. Adding a single attention layer significantly improves performance, and further gains are observed with a 1-layer transformer encoder. The 2-layer transformer offers marginal additional benefit, suggesting that a lightweight transformer (1–2 layers) is sufficient for effective hyper-token modeling.

Table 8: **Ablation of hyper-encoder architecture** on byte-perplexity ($\downarrow$) across four corpora using a 1024-token context window. Performance improves with increasingly expressive architectures.

| Model | Method | Wiki | Pile | mC4 | dC4 |
|-------|--------|------|------|-----|-----|
| Phi-3.5-4B | averaging | 1.81 | 1.97 | 2.29 | 2.08 |
| | 1-attention-layer | 1.73 | 1.86 | 2.16 | 2.01 |
| | 1-transformer-layer | 1.71 | 1.83 | 2.13 | 1.99 |
| | 2-transformer-layer | 1.72 | 1.84 | 2.15 | 2.00 |

Figure 6 illustrates the effect of hyper-encoder architecture on zip2zip training loss. We observe that the simple averaging method converges the fastest but plateaus at a relatively high loss, reflecting its limited capacity. As model complexity increases—with attention and transformer layers—the convergence becomes slower, yet the final loss is significantly lower. Notably, the 1-layer and 2-layer transformer encoders yield the best performance, demonstrating that additional parameters enable the model to better capture structure, albeit at the cost of slower training dynamics.

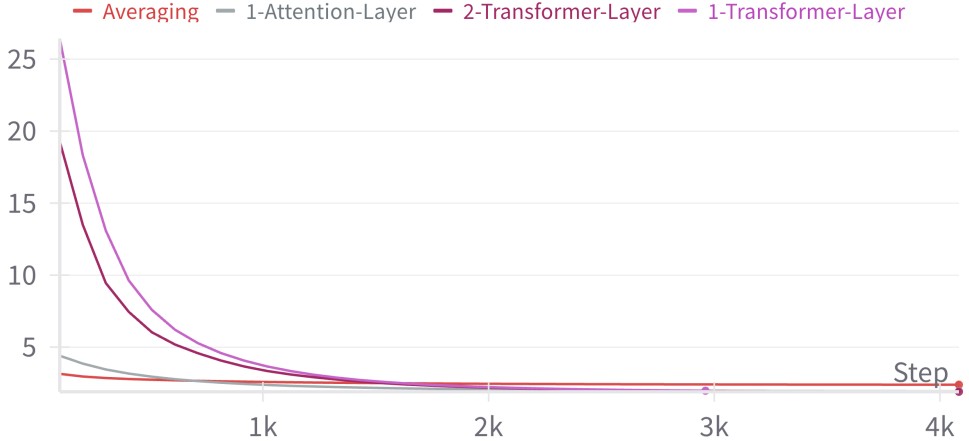

Figure 6: **Effect of hyper-encoder architecture on zip2zip training loss.** Averaging (no additional parameters) converges quickly but to a higher loss. As architectural complexity increases—from attention to transformer layers—convergence becomes slower, but the final loss is lower. This highlights a trade-off between training speed and modeling capacity.

## B. FLOPs Estimation for `zip2zip`

Following the assumptions of Kaplan et al. (2020), we estimate training FLOPs ($\Gamma$) as:

$$\Gamma \approx 6 \cdot N_{\text{tokens}} \cdot N_{\text{params}},$$

where $N_{\text{tokens}}$ is the total number of processed tokens and $N_{\text{params}}$ is the number of trainable parameters. This estimate ignores the quadratic attention cost, assuming:

$$12 \cdot d_{\text{model}} \ll \text{sequence length.}$$

For `zip2zip`, this becomes:

$$\Gamma_{\text{z2z}} \approx 6 \cdot N_{\text{tokens}} \cdot \rho \cdot N_{\text{params}}(1 + \alpha),$$

where $\rho$ is the compression ratio, and $\alpha$ accounts for the overhead of the hyper-encoder applied at the embedding and LM head. The relative FLOPs ratio is then:

$$\frac{\Gamma_{\text{z2z}}}{\Gamma} = \rho \cdot (1 + \alpha).$$

Assuming the hyper-encoder mirrors the base model's configuration, we estimate:

$$\alpha \approx \frac{lM}{L},$$

where $l$ is the number of hyper-encoder layers, $M$ is the maximum merge size, and $L$ is the number of base model layers. We illustrate this estimate across several model scales in Table 9, showing that the relative FLOPs overhead from the hyper-module remains modest (typically under 15%).

| Model | L | M | l | $\alpha = \frac{lM}{L}$ |
|---|---|---|---|---|
| LLM-4B | 14 | 2 | 1 | 0.14 |
| LLM-7B | 32 | 2 | 2 | 0.13 |
| LLM-70B | 80 | 3 | 3 | 0.11 |
| LLM-400B | 128 | 3 | 4 | 0.09 |

Table 9: Relative FLOPs overhead from the hyper-module across different model sizes.

## C. Additional Results

**Machine Translation**

We report standard deviations for machine translation results across WMT benchmarks in Table 10, computed using the lm-evaluation-harness codebase.

Table 10: Machine translation performance on WMT benchmarks (BLEU↑, CHRF↑, TER↓) with standard deviations (±) from bootstrapped estimates. Scores are averaged across both directions.

| Model | Method | WMT14 En-Fr | | | WMT16 En-De | | | WMT16 En-Ro | | |
|---|---|---|---|---|---|---|---|---|---|---|
| | | BLEU | CHRF | TER | BLEU | CHRF | TER | BLEU | CHRF | TER |
| Phi-3.5-4B | Base | 33.6±2.1 | 58.3±1.4 | 53.0±1.7 | 39.2±1.9 | 63.2±1.6 | 47.9±1.8 | 17.7±1.5 | 45.5±1.3 | 73.4±2.4 |
| | Cont. finetune | 36.5±2.2 | 61.0±1.6 | 51.5±1.8 | 42.3±1.8 | 65.4±1.4 | 44.9±1.7 | 16.7±1.4 | 45.8±1.5 | 79.7±2.3 |
| | zip2zip | 34.1±1.9 | 59.4±1.5 | 54.5±2.0 | 39.7±1.7 | 64.5±1.6 | 48.0±1.9 | 14.3±1.6 | 44.2±1.4 | 93.5±2.5 |
| Phi-3-14B | Base | 39.1±2.0 | 62.6±1.4 | 49.3±1.9 | 43.1±2.0 | 65.6±1.5 | 44.1±1.7 | 21.3±1.5 | 51.0±1.4 | 70.5±2.2 |
| | Cont. finetune | 38.9±2.2 | 63.2±1.4 | 48.8±1.9 | 48.4±2.0 | 70.1±1.3 | 39.8±1.9 | 21.8±1.4 | 52.0±1.3 | 68.3±2.9 |
| | zip2zip | 36.4±2.1 | 62.8±1.5 | 51.2±1.8 | 44.8±2.1 | 68.1±1.6 | 42.9±1.8 | 19.5±1.5 | 50.1±1.3 | 72.9±2.6 |

## D. Technical Details

**Model and Training Configuration**

- **Pretrained Model:** `microsoft/Phi-3-medium-4k-instruct`

- **Sequence Length:** 1024

- **Total Batch Size:** 32,768 tokens

- **Learning Rate Schedule:** Cosine decay

- **Learning Rate Range:** Max = 3e-4, Min = 1e-5

- **LoRA rank and alpha value:** Both are 32

- **Training Steps:** 10,000

- **Validation Interval:** Every 100 steps

- **Checkpoint Interval:** Every 500 steps

- **Pytorch Model Compilation:** Enabled

**LoRA Configuration**

- **Rank:** 16

- **Alpha:** 16

- **Target Modules:** qkv_proj, o_proj, gate_proj, down_proj, up_proj

**System and Libraries**

- **Hardware:** 4 × NVIDIA A100-SXM4-80GB GPUs, 64-core CPU (128 threads)

- **Key Libraries:**

  - PyTorch >= 2.5.0
  - Transformers >= 4.47.0
  - Datasets <= 3.1.0
  - Accelerate >= 0.26.0

**Compute Resources**

We report the compute resources used for training our models in Table 11. All training was conducted on internal servers equipped with NVIDIA H100 GPUs. We estimate GPU-hours by multiplying wall-clock training time by the number of GPUs used. No additional compute was used beyond the reported experiments; we did not perform parameter grid search, large-scale hyperparameter tuning, or exploratory runs that were excluded from the paper.

Table 11: Training compute resources for zip2zip experiments.

| Model | GPUs | Time | GPU Type | GPU-Hours |
|---|---|---|---|---|
| Phi-3.5-Medium (14B) | 4 | 15h 46m | NVIDIA H100 80GB | 63.0 |
| Phi-3.5-Mini (4B) | 2 | 7h 0m | NVIDIA H100 80GB | 14.0 |

**Inference.**  All evaluations complete within 1 hour on a single A100 GPU, demonstrating the runtime efficiency of zip2zip.

## E. Data Mixture

To support effective fine-tuning, we construct a curated dataset with balanced representation across diverse domains, including code, mathematics, dialogue, general web content, and multilingual text. The final dataset contains approximately 1 billion compressed tokens.

Table 12 summarizes the constituent datasets and their respective proportions. A visualization of the dataset composition and sequence length characteristics is shown in Figure 7.

The multilingual subset in fineweb-2 includes the following languages: Mandarin Chinese (cmn_Hani), German (deu_-Latn), Japanese (jpn_Jpan), Spanish (spa_Latn), French (fra_Latn), Italian (ita_Latn), Portuguese (por_Latn), Dutch (nld_Latn), and Arabic (arb_Arab).

| Dataset | Domain | Proportion (%) |
|---|---|---|
| HuggingFaceFW/fineweb-edu(Lozhkov et al., 2024a) | Web / Knowledge | 20% |
| devngho/the-stack-llm-annotations-v2(Lozhkov et al., 2024b) | Code | 25% |
| AI-MO/NuminaMath-1.5(LI et al., 2024) | Math | 20% |
| HuggingFaceH4/ultrachat_200k(Ding et al., 2023) | Chat / Dialogue | 20% |
| HuggingFaceFW/fineweb-2(Penedo et al., 2024) | Multilingual | 15% |

Table 12: Training data composition across domains.

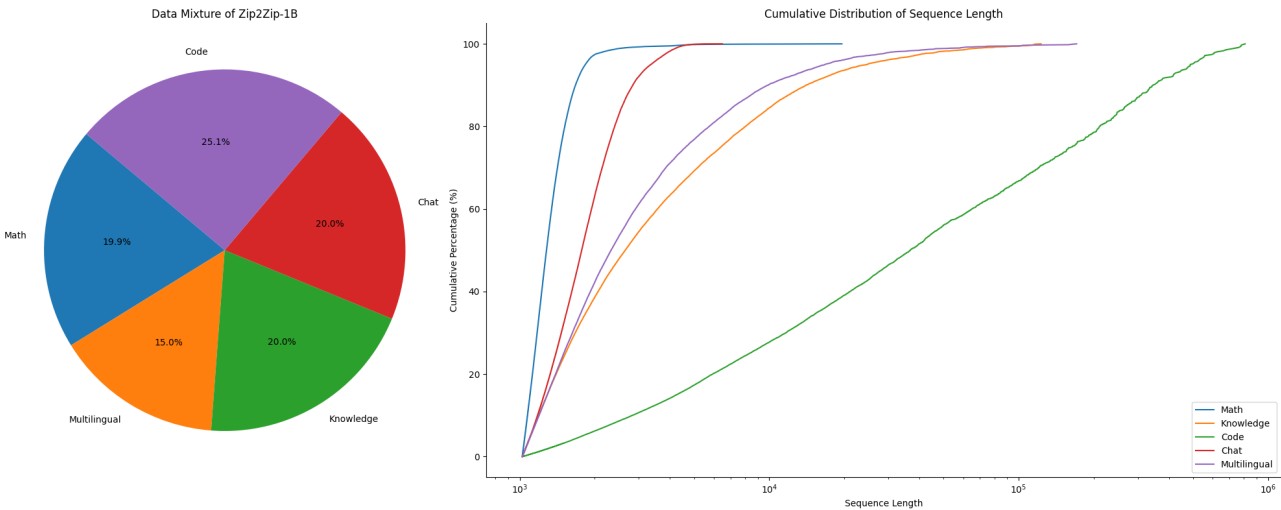

Figure 7: Left: Proportional breakdown of the fine-tuning dataset across five domains. Right: Cumulative distribution of input sequence lengths per domain (log scale). Code and multilingual data exhibit longer tail distributions, indicating greater variability in sequence lengths.

# F. Token Stream Visualization

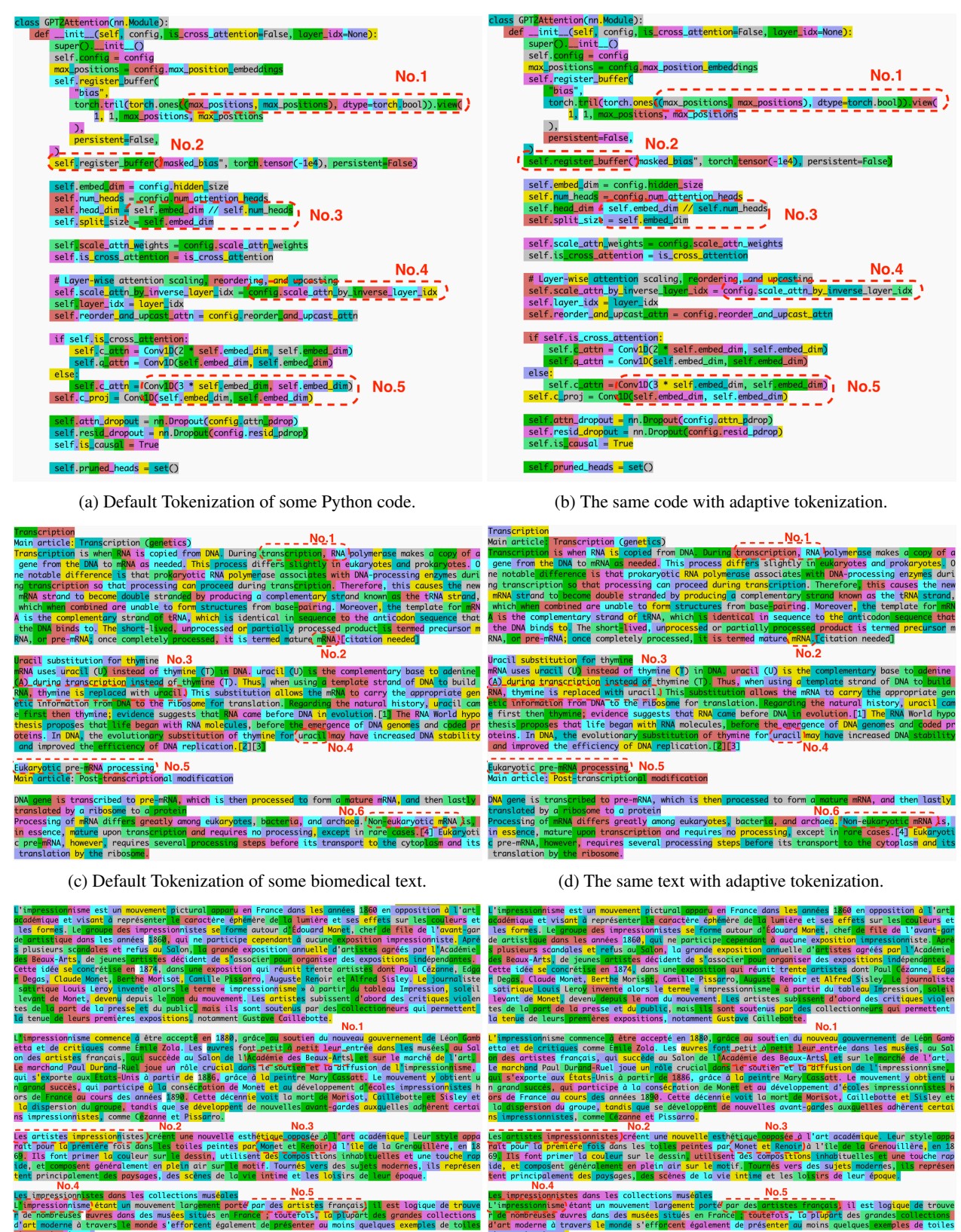

(a) Default Tokenization of some Python code.

(b) The same code with adaptive tokenization.

(c) Default Tokenization of some biomedical text.

(d) The same text with adaptive tokenization.

(e) Default Tokenization of text in French.

(f) The same text with adaptive tokenization.

Figure 8: Examples comparing default and adaptive tokenization. Dotted-line frames highlight where the differences are most noticeable.

