# OpenReview forum: "zip2zip: Inference-Time Adaptive Vocabularies for Language Models via Token Compression"
_ICML.cc/2025/Workshop/TokShop — TokShop_

### Official Review · Reviewer_AWYM · 2025-06-06
**Fairly novel approach with evaluation that can be improved by including a few more details**

**Rating:** 6
**Confidence:** 3

**Review:**

The authors present their method for inference-time adaptive vocabulary expansion using the LZW compression algorithm together with an auxiliary transformer network for merging tokens and their embeddings. They demonstrate that the proposed compression method results in shorter output sequence length and lower inference time while maintaining similar or slightly worse MT performance and NLP benchmark accuracy when compared to the baseline without the token compression. Still, a more meaningful comparison should also include other competing compression methods such as the BLT mentioned later in the paper.
The authors also present a brief summary of the limitations of their approach.

I still have a few questions and suggestions:

It is not clear to me, how exactly is the hyper-encoder (e.g. 2-layer transformer) used to generate the output hyper-tokens since I would expect that you would need to invert the function computed by the whole hyper-encoder transformer. Can you please add a more detailed description of the hyper-token output projection?

In related work section, you mention the compression using BLT and compare it to your approach. It would be interesting to add an experimental comparison between the said method and your approach.

If I did not miss anything, it seems that you only evaluated your method on the languages with latin script. It would be interesting how effective your method can handle cases with multiple different writing scripts (nowadays an increasingly important application scenario).

I suggest including the original scores (and not just the standard deviation) in Table 10 for easier reading.

There is a missing Table (??) reference on pages 7 and 9. Please fix that.

---

### Official Review · Reviewer_dETY · 2025-06-09
**This is a token compression work with a good design and promising results, yet it also acknowledges its limitations honestly.**

**Rating:** 8
**Confidence:** 4

**Review:**

The authors propsoe zip2zip, a framework that uses LZW compression to dynamically shorten token sequences during inference, thereby increasing speed. By finetuning existing models w/ adaptors, it achieves significant latency reductions. However, this speed comes at the cost of notable performance degradation, particularly on tasks requiring precision.

Strong points:
S1: The core idea of inference-time vocabulary adaptation is a novel and practical approach to a known problem.

S2: The efficiency evaluation is thorough, providing real-world latency metrics on relevant hardware that demonstrate significant speedups.

S3: The ablation studies on merge size and hyper-encoder architecture are well-executed and validate key design choices.


Weak points:
W1: The catastrophic performance drop on GSM8K (Phi-3-14B, from 84% to 25% accuracy) renders the method unusable for any task requiring numerical or structural precision.

W2: The framework consistently increases byte-level perplexity and lowers scores on most NLP benchmarks, indicating a fundamental trade-off of quality for speed.

W3: The mechanism for controlling compression—a single "maximum merge size" hyperparameter—is primitive, and its effects are shown to be unstable.

Minor points:
Line 347, 448: Table ??

---

### Decision · Program_Chairs · 2025-06-10

Accept